# Plastid Phylogenomics and Plastomic Diversity of the Extant Lycophytes

**DOI:** 10.3390/genes13071280

**Published:** 2022-07-19

**Authors:** Sisi Chen, Ting Wang, Jiangping Shu, Qiaoping Xiang, Tuo Yang, Xianchun Zhang, Yuehong Yan

**Affiliations:** 1Shenzhen Key Laboratory for Orchid Conservation and Utilization, National Orchid Conservation Center of China and the Orchid Conservation & Research Center of Shenzhen, Shenzhen 518114, China; 17770972237@163.com (S.C.); tingwang93@126.com (T.W.); jpshu@scbg.ac.cn (J.S.); yangtuo416@outlook.com (T.Y.); 2State Key Laboratory of Systematic and Evolutionary Botany, Institute of Botany, Chinese Academy of Science, Beijing 100093, China; qpxiang@ibcas.ac.cn; 3University of Chinese Academy of Sciences, Beijing 100049, China; 4College of Biodiversity Conservation, Southwest Forestry University, Kunming 650224, China; 5Key Laboratory of Plant Resources Conservation and Sustainable Utilization, South China Botanical Garden, Chinese Academy of Sciences, Guangzhou 510650, China

**Keywords:** lycophytes, plastid genome, comparative genomics, clubmoss, plastomic diversity

## Abstract

Although extant lycophytes represent the most ancient surviving lineage of early vascular plants, their plastomic diversity has long been neglected. The ancient evolutionary history and distinct genetic diversity patterns of the three lycophyte families, each with its own characteristics, provide an ideal opportunity to investigate the interfamilial relationships of lycophytes and their associated patterns of evolution. To compensate for the lack of data on Lycopodiaceae, we sequenced and assembled 14 new plastid genomes (plastomes). Combined with other lycophyte plastomes available online, we reconstructed the phylogenetic relationships of the extant lycophytes based on 93 plastomes. We analyzed, traced, and compared the plastomic diversity and divergence of the three lycophyte families (Isoëtaceae, Lycopodiaceae, and Selaginellaceae) in terms of plastomic diversity by comparing their plastome sizes, GC contents, substitution rates, structural rearrangements, divergence times, ancestral states, RNA editings, and gene losses. Comparative analysis of plastid phylogenomics and plastomic diversity of three lycophyte families will set a foundation for further studies in biology and evolution in lycophytes and therefore in vascular plants.

## 1. Introduction

Vascular plants evolved from ancestral vascular-free land plants more than 400 million years ago [1,2], which is a major step during the colonization of land by plants [3,4]. As a sister group of all other extant terrestrial plants and the living clade most similar to early vascular plants of the fossil record, lycophytes retain many typical features of early vascular plants [5], and hence are suitable for investigating the early evolution of vascular plants. Although they dominated the earth’s flora through the Devonian and Carboniferous periods [6,7], only three distinctive lineages (clubmosses, quillworts, and spikemosses) are extant, corresponding to three families: Lycopodiaceae, Isoëtaceae, and Selaginellaceae [8]. There are perhaps 700 species in Selaginellaceae within a single genus, about 388 species in Lycopodiaceae in 16 genera, and approximately 250 species in the single monogeneric family Isoëtaceae [8]. Their early divergence and evolutionary innovations enable their broad global distribution and diverse ecological niches, as well as abundant diversity at the genetic level [9].

Plastids are vital organelles for photosynthesis, amino acid and lipid synthesis, and many other biochemical pathways in plants. They provide energy to green plants and play a vital role in the growth and development of plants. They are usually uniparentally inherited, conserved in structure, and easy to be sequenced for having a high copy number, thus being an ideal choice for plant phylogenetics and evolutionary inquiry [10]. The advent of next-generation sequencing techniques and open data sources makes it much easier for researchers to sequence, download, and analyze complete plastid genomes (plastomes), and with more and more plastome data available in databases, universal features of the plant plastome have been gradually revealed. Plastomes of most organisms have a quadripartite circular structure, ranging from 115 to 165 Kb in size [11,12]. In general, a typical quadripartite plastome consists of two inverted repeat (IR) sequences: a large single copy (LSC) and a small single copy (SSC). However, some exceptions exist. Loss of one of the IR regions is not rare in some clades of seed plants, such as Pinaceae [13], cupressophytes [14], Passiflora [15], and some clades of Fabaceae [16]. In lycophytes, a pair of direct repeats (DR) was found in most species of Selaginellaceae [17], while Lycopodiaceae and Isoëtaceae share typical IR structures [18]. As for other characteristics of plastomes, such as GC content, organelle genomes are near universally rich in AT throughout the eukaryotic domain, with an average GC content of 45% [19]. The highest plastid GC content holding the record appears in *Selaginella*
*remotifolia*, at 56.5% [17], and the GC content of nearly all Selaginellaceae plastomes is extraordinarily high. Interestingly, despite their common ancient origins, the plastomes of the other two lycophyte families are much lower in GC content. As reported in previous studies, the GC content of Selaginellaceae plastomes ranges from 50.7% in *Selaginella lyalliito* to 56.5% in *S. remotifolia* [17]. However, the GC contents of Lycopodiaceae and Isoëtaceae are relatively low when compared throughout all ferns and lycophytes [20]: the GC content of Lycopodiaceae ranges from 35% to 36%, that of Isoëtaceae is approximately 38% [18], and that of ferns ranges from 33% to 45% in plastids. The plastid structure and features of the different families of the lycophytes differentiated in distinctive directions, becoming an intriguing evolutionary puzzle that has attracted researchers for a long time [17,18,21,22,23].

The exponential increase in the speed and accuracy of plastid sequencing has contributed to our understanding of lycophytes. To date, many DNA fragments have been applied to research on lycophytes [24,25,26,27,28,29], and since the publication of the first lycophyte plastome of *Huperzia lucidula* (Michx.) Trevis. (Lycopodiaceae) in 2005 [30], remarkable achievements have been made in the molecular evolution and classification of lycophytes. From 14 plastids reported in 2018 [18] to 81 complete lycophyte plastomes in this article (14 newly sequenced), combined with published nuclear genomes of *Selaginella kraussiana* (Kunze) A. Braun [5], *S. moellendorffii* Hieron. [31], *S. tamariscina* (P. Beauv.) Spring [32], and *Isoëtes taiwanensis* De Vol [33], numerous mysteries of lycophytes have gradually been uncovered in recent years.

However, although several studies on lycophyte plastomes have been reported to date, they have covered relatively few species and have focused on only limited genera or families, resulting in an insufficient understanding of the lycophytes as a whole [17,18,21,22,34,35,36]. Indeed, compared with those of other terrestrial plants, the plastomes of lycophytes are relatively scarce, particularly for Lycopodiaceae. To explore the evolutionary patterns and strategies of the entire lycophyte lineage and the differences among the three lycophyte families, more samples and more species must be sequenced and investigated.

In this study, 14 plastomes were newly sequenced and assembled to compensate for the lack of data on Lycopodiaceae, thus enriching the plastome data of the lycophytes. The new plastomes were combined with 67 previously published lycophyte plastomes and 12 plastomes of other lineages to perform a comprehensive plastomic analysis of the lycophytes. In total, we analyzed 93 plastomes and aimed to (i) reconstruct the lycophyte phylogeny based on plastid protein-coding sequences; (ii) present the global structural characteristics of plastids from the three lycophyte families (Lycopodiaceae, Isoëtaceae, and Selaginellaceae); and (iii) characterize the features, differences, and evolutionary trajectory of lycophyte plastomes by comparing plastomic diversity of the different families in terms of GC content, variation of gene number, divergence time, substitution rate, RNA editing.

## 2. Materials and Methods

### 2.1. Taxon Sampling, DNA Extraction, and Sequencing

Fourteen individuals from 12 species, *Lycopodiastrum casuarinoides* (collection No. YYH15707, from Nanling, China), *Diphasiastrum complanatum* (YYH15882, from Bijie, China), *Lycopodium japonicum* (YYH15880, from Bijie, China), *Dendrolycopodium obscurum* (YYH15871, from Changbai Mountain, China), *Dendrolycopodium verticale* (YYH15881, from Bijie, China), *Pseudolycopodiella caroliniana* (YYH15570, from Yangjiang, China), *Palhinhaea cernua* (YYH15722 from Nanling, China; css1 from Shenzhen, China; YYH15886, from Jieyang, China), *Huperzia crispata* (146, from Hongya, China), *Huperzia selago* (3239, from Eastern Alps, Austria), *Phlegmariurus henryi* (YYH15875, from Fangchenggang, China), *Phlegmariurus mingcheensi* (YYH15514, from Dawuling Natural Reserve, China), and *Phlegmariurus squarrosus* (YYH15879, from Jingxi, China), were used for genome skimming and complete plastome assembly (Appendix A). All leaf samples were frozen in liquid nitrogen and stored at −80 °C. Total DNA was extracted from the leaves using a modified CTAB method and sequenced using short reads produced by the NovaSeq 6000 platform (2 × 150 bp) by Novogene (Beijing, China).

### 2.2. Plastome Assembly, and Annotation

The raw reads were filtered using Trimmomatic 0.39 [37] by removing low-quality bases and sites, and the parameters were set as LEADING = 20, TRAILING = 20, SLIDINGWINDOW = 4:15, MINLEN = 36, and AVG QUAL = 20. The quality was assessed using FastQC 0.11.9 [38]. The filtered reads were assembled using the pipelines GetOrganelle 1.6.2 [39] with kmers: 21,45,65,85,105. Bandage 0.8.1 [40] was then used to visualize and complete the assembly. The annotation of plastomes was performed using PGA [41] and then visually inspected and edited by hand where necessary in Geneious v11.0.4 [42]. Published lycophyte plastomes (*Diphasiastrum digitatum*: MH549638, *D. obscurum*: MH549637, *Lycopodium clavatum*: MH549642, *H. lucidula*: AY660566 and MH549639, *Huperzia serrata*: KX426071, *H. javanica*: KY609860, *Phlegmariurus carinatus*: MN566837, *Phlegmariurus phlegmaria*: MT786212) are used as references in all assembly and annotation analysis.

### 2.3. Phylogenetic Analysis

This study included one genus of Isoëtaceae, one genus of Selaginellaceae, and eight genera representing all three subfamilies (Huperzioideae, Lycopodioideae, and Lycopodielloideae) of Lycopodiaceae. In addition to the 14 newly sequenced Lycopodiaceae plastomes, we downloaded the GenBank and Fasta format files for plastomes from 67 other lycophytes (31 Isoëtaceae, 27 Selaginellaceae, and 9 Lycopodiaceae) and 12 outgroup species (3 each of Bryophyta, Gymnospermae, Angiospermae, and Pteridophyta) from NCBI. GenBank accessions and more extensive data are listed in Appendix A. For plastomes with incomplete or no annotations, PGA was used for annotation, and Geneious 11.0.4 was used for visual inspection, with previously reported, closely related plastomes as references.

Phylogenetic analyses were conducted on 93 plastomes. First, all coding sequences (CDSs) were extracted in batches using the Python script get_annotated_regions_from_gb.py (https://github.com/Kinggerm/PersonalUtilities/, last accessed on 12 July 2022) [43]. A total of 84 CDSs with over 50% species coverage were selected. All sequences were aligned using Mafft 7.487 [44] and trimmed with Gblocks 0.91b [45,46], with default parameters to remove poorly aligned regions. Finally, we concatenated all CDSs into one matrix and partitioned them by codon position (1st, 2nd, and 3rd) in PhyloSuite 1.2.2 [47]. The best partitioning scheme and substitution model of the matrix was determined in PartitionFinder 2.0.0 [48] under the Bayesian information criterion (BIC), using the relaxed hierarchical clustering algorithm [49], and with the branch length linked. Based on the best partition scheme suggested by PartitionFinder, both Bayesian inference (BI) and maximum likelihood (ML) methods were applied in phylogenetic inference. A maximum likelihood (ML) tree was built using IQ-TREE 2.1.4 [50] with 1000 bootstrap replicates. Another phylogenetic tree was constructed with MrBayes 3.2.7 [51]; two simultaneous runs were performed, each consisting of four chains, and the parameters were set to “lset nst = 6 rates = invgamma” for Bayesian inference (BI) analysis. In total, chains were run for 3,000,000 generations, with topologies sampled every 1000 generations. The first 25% of the sampled trees were discarded as burn-in. The convergence of runs was assumed when the average standard deviation of split frequencies dropped below 0.01. The best tree was edited using FigTree v1.4.4 (http://tree.bio.ed.ac.uk, last accessed on 12 July 2022), with ML bootstrap (BS) and Bayesian posterior probability (PP) support values shown for each node (Figure 1).

### 2.4. Comparative Plastomic Analysis

To focus on the evolutionary trajectories of the lycophytes, comparative plastomic analyses were performed on the lycophyte plastomes. The basic characteristics of lycophyte plastomes, such as genome length and GC content of overall, single copy, and long repeat regions, are listed in Appendix A. To compare the structure and gene contents of the plastomes, one of the repeat regions was excluded, and pairwise genome alignments among the plastomes were performed separately for each family using progressive Mauve 2.4.0 [52].

The GC content along the complete sequence of each plastome was calculated with a sliding window size of 100 bp. Taking into account the particularity of repeat regions, separate sliding window detection was performed for repeat regions using the same sliding window size. Finally, eight widespread and representative species were selected for display because they could represent the common variation in each clade, and they are wildly distributed and well-known species (*H. serrata*, *H. selago*, *P. phlegmaria*, *P. cernua*, *L. japonicum*, *Isoëtes sinensis*, *Selaginella tamariscina*, and *S. lepidophylla*). Figures were prepared using the R language, with annotation information for each plastome obtained from Geneious 11.0.4 and depicted at the bottom of each scatter plot.

### 2.5. Divergence Time Estimation and Ancestral State Reconstruction

Before divergence time estimation and ancient trait reconstruction, we deleted all duplicate species, subspecies, varieties, forms, and species affinis, and kept only one plastome for each species. Finally, 71 plastomes from 93 plastomes were selected for analysis. With the same software and parameters described in Section 2.3, 84 CDSs were selected, aligned, and pruned into alignments. These alignments were concatenated and partitioned by codon position (1st, 2nd, and 3rd) in PhyloSuite 1.2.2 [47]. After the best partitioning scheme and substitution model were determined in PartitionFinder 2.0.0 [48], maximum likelihood (ML) inferences were performed using IQ-TREE 2.1.4 [50] with 1000 bootstrap replicates. This ML tree was used for divergence-time estimation and ancient trait reconstruction.

We estimated divergence times of the lycophytes with the MCMCTree program implemented in PAML 4.9 [53] based on the partitioned CDS dataset (partitioned by 1st, 2nd, and 3rd codon positions) and the species tree reconstructed above. MCMCtreeR was used to prepare the MCMCtree analyses and visualize the time-scaled trees [54]. Four fossils were used for calibration, following [2] and [36]. The divergence between Isoëtales and Selaginellales was estimated to be 331–388 Ma, based on *Yuguangia ordinata* [55]. The node marking the divergence of heterosporous lycophytes (Isoëtaceae plus Selaginellaceae) and homosporous lycophytes (Lycopodiaceae) was constrained at 392–451 Ma based on *Leclercqia complexa* [56]; the crown node of Euphyllophyta was constrained at 386–451 Ma based on *Rellimia thomsonii* [57] and the Embryophyta crown node was estimated at 469–516 Ma based on *Tetrahedraletes* cf. *medinensis* [58], as proposed by Morris et al. [2]. Uniform distributions were applied to specify the prior distributions on node ages, allowing 0.001% probability of an age younger or older than the given minima and maxima. The time unit was set to 100 Ma. The clock and the model in the control file were set as independent rates and the GTR + G model (model = 7), respectively. The Markov chain Monte Carlo (MCMC) chains were run for 200 million generations, sampling every 100 generations, and the first 20 million generations were discarded as burn-in. Two independent runs were performed, and the stationary state and convergence of each run were assessed in Tracer 1.7.2 [59] to ensure sufficient and effective sample sizes for all parameters surpassing 200.

Ancestral trait reconstruction was performed to investigate how the gene number and GC content evolved across the lycophyte phylogeny. Ancestral GC contents and gene numbers were reconstructed using the function “ace” from the R package ape based on maximum likelihood (ML) methods on continuous characters [60] and mapped out through the corresponding branches of the divergence tree by the *ggtree* package in R [61].

### 2.6. Estimation of Substitution Rates

To compare substitution rates in the three families, the overall substitution rates (OSRs) of every ribosomal RNA (rRNA) gene and two noncoding region datasets (IR dataset and SC dataset) were estimated with BASEML in PAML 4.9 [53] using the GTR (REV) substitution model. For other genes, a concatenated matrix of all CDSs was used to estimate the synonymous substitution rate (dS) and nonsynonymous substitution rate (dN) with the codeml program in PAML 4.9 under the branch model (model = 2). Three branches (corresponding to the three families) were regarded as having three separate ω (dN/dS) values, but the same ω value was shared within every family, and each “foreground” and “background” branch was specified. For the resulting OSR trees, dN trees, and dS trees, the branch lengths from the root to each tip were calculated using the R package castor 1.6.1 [62].

### 2.7. RNA Editing Analysis

A fraction of RNA editing would alter the sequences of start codons and stop codons, thus providing an opportunity for RNA editing inferences with a lack of mature mRNA or cDNA. To estimate the number of RNA-editing sites in each lycophyte plastome, the GFF3, GenBank, and Fasta files of all CDSs from each plastome were extracted separately, and the number and type of potential RNA-editing sites in the stop codons were then counted using the R package ReFernment [63]. ReFernment can automatically detect apparent nonsense mutations in plastid genomes to annotate hypothesized RNA-editing sites. Although it detects fewer RNA-editing sites than other similar tools, such as PREPACT, its results are more reliable and suitable for this research. Because some plastomes were published with the RNA editing corrected before uploading them to NCBI [64], the RNA-editing sites of a plastome that were obviously atypical compared with those of other species in the same family were treated as outliers and removed. For two of the outliers in Isoëtaceae, *Isoëtes hypsophila* and *Isoëtes japonica* [64], we contacted the authors and obtained the original plastome sequences.

## 3. Results

### 3.1. Features of Lycophyte Plastomes

Fourteen new plastomes from 12 species of Lycopodiaceae were sequenced, assembled, and annotated in this study. Detailed information on these plastomes is listed in Appendix A, together with information on the other 79 downloaded plastomes. These newly sequenced plastomes exhibit a typical quadripartite structure: a large single copy region (LSC), a short single copy region (SSC), and two inverted repeats (IRs). The lengths of these complete plastomes range from 145,398 bp (*P. cernua*) to 168,075 bp (*Huperzia selago*), and their overall GC contents range from 32.9% (*L. casuarinoides*) to 36.3% (*Huperzia crispata*).

There are obvious differences in the plastomes of the three families. In terms of GC content, that of Lycopodiaceae was the lowest (32.9–36.4%), followed by Isoëtaceae (37.7–38.2%) and Selaginellaceae (50.2–56.5%). In terms of plastome size, those of Selaginellaceae are the shortest, from 110,164 bp (*Selaginella nipponica*) to 148,924 bp (*S. sanguinolenta*), followed by those of Isoëtaceae, from 142,880 bp (*Isoëtes* aff. *amazonica*) to 146,362 bp (*I. hypsophila*), and Lycopodiaceae, from 145,297 bp (*P. cernua*) to 168,075 bp (*Huperzia selago*). A reduction in genome size was associated with a reduction in gene number. Selaginellaceae plastomes have the smallest number of genes, with the fewest in *Selaginella pallidissima* (64 genes) and the most in *S. sanguinolenta* (113 genes), followed by Isoëtaceae (129 genes on average), and Lycopodiaceae (131 genes on average) (Appendix A). In general, Selaginellaceae plastomes had the lowest gene numbers, the shortest lengths, and the highest GC contents; Lycopodiaceae plastomes had the largest gene numbers, the greatest lengths, and the lowest GC contents; and the Isoëtaceae plastomes were intermediate.

Plastome rearrangement analysis was performed by family, and the results show that there has been little genome rearrangement in Lycopodiaceae and Isoëtaceae (Appendix A), especially Isoëtaceae, which showed almost no structural rearrangement or extreme structural conservation (Appendix A). However, significant structural rearrangements appeared in Selaginellaceae (Appendix A). Unlike Selaginellaceae, the other two families show normal IR structures.

### 3.2. Phylogenetic Analyses

A robust phylogenetic tree of lycophytes was constructed based on 81 plastomes from lycophytes and 12 from other land plants (Figure 1). In our phylogenomic analyses, Lycophytes are sister to all other extant vascular plants and can be divided into three clades that correspond to three families (Lycopodiaceae, Isoëtaceae, and Selaginellaceae). BI and ML analyses recovered highly supported tree topologies, with the shortest branch lengths in Isoëtaceae, intermediate in Lycopodiaceae, and longest in Selaginellaceae (Figure 1). The phylogeny reconstructed the topology of eight genera of Lycopodiaceae: *Huperzia*, *Phlegmariurus*, *Dendrolycopodium*, *Diphasiastrum*, *Palhinhaea*, *Pseudolycopodiella*, *Lycopodiastrum*, and *Lycopodium*. However, the topology of the genus *Dendrolycopodium* is slightly discordant between the ML tree and the BI tree. In the BI tree, the genus *Diphasiastrum* sisters to *Dendrolycopodium*, while *Diphasiastrum* sisters to *Lycopodium* instead in the ML tree, both with low support values (0.64 for BI and 53 for ML) (Figure 1). As for Isoëtaceae and Selaginellaceae, their topology was divided into two lineages. 

### 3.3. GC Content Distribution among Plastomes

To investigate the specific location of GC differences on different plastomes, we performed a sliding-window analysis of GC content in each plastome and detected GC peaks in rRNA genes and their adjacent regions. The GC distributions along eight representative plastomes are shown in Figure 2. In Isoëtaceae and Lycopodiaceae, the GC content was highest in the repeat regions, with two GC peaks in the IR regions, as shown in the middle panels in Figure 2. These peaks clearly corresponded to regions containing rRNA genes. The GC content of the repeat regions was slightly higher in Selaginellaceae than in the six Isoëtaceae and Lycopodiaceae plastomes in Figure 2 (Appendix A), but in Selaginellaceae, the two GC peaks were not as prominent as in Isoëtaceae and Lycopodiaceae, due to the smaller GC difference between repeat regions and single copy regions. *Selaginella tamariscina*, with the DR structure, and *S. lepidophylla*, with the IR structure, shared a similar pattern in which the GC contents of the SC region and repeat region were nearly identical, regardless of the order of the two repeats. In conclusion, as well as an increase in the GC content of the repeat regions, Selaginellaceae showed an increase in the GC content of the SC region to a level similar to, though still not as high as that of the two repeat regions (Appendix A). In contrast, the GC content of the IR regions was much higher than that of the SC region in the two other families of lycophytes. 

### 3.4. Base Substitution Rate Estimation

As shown in Figure 3, the mutation rates of each family showed entirely different patterns. Within each family, the nonsynonymous substitution rate (dN) is always lower than the synonymous substitution rate (dS), and the coding regions always mutate more slowly than the noncoding regions. Among the three families, whether for dN or dS, in noncoding regions (OSR) or coding regions, and in repeat or single-copy regions, the substitution rate is generally much higher in Selaginellaceae than in Lycopodiaceae and Isoëtaceae. In addition, the substitution rates in the SC and IR regions increased synchronously, with substitution rates always lower in the two repeat regions than in the SC region. rRNA genes have extremely low substitution rates and extremely high GC contents, and peaks in GC content in the IR regions of plastomes are clearly caused by the presence of these genes (Figure 2). These results indicate low, intermediate, and high rates of plastomic change in the plastids of Isoëtaceae, Lycopodiaceae, and Selaginellaceae, respectively.

### 3.5. RNA Editing Analysis

In this study, we predicted only the numbers and types of RNA editing in CDSs. Among the three families, Selaginellaceae showed higher levels of C-to-U RNA editing (Figure 4). Plastomes of Lycopodiaceae contain both C-to-U and U-to-C RNA-editing sites, and the latter are far more common. Two completely different patterns appear in Isoëtaceae: some exhibit almost no C-to-U RNA editing, while others show both types of editing, with C-to-U predominating. The Selaginellaceae is estimated to have the most extensive C-to-U RNA editing numbers. Followed by Isoëtaceae splitting into two clades, both have similar U-to-C editing numbers, but one clade exhibits elevated C-to-U editing sites. Lycopodiaceae has the lowest number of RNA editing sites.

### 3.6. Divergence Time Inference and Ancestral State Reconstruction

The divergence time inference and the ancestral state reconstruction of GC content and gene number in the lycophytes are shown in Figure 5. Based on the four calibration points derived from fossils, our analysis estimated that the stem age of the lycophytes was dated to 417 Ma in the Lower Devonian, with a 95% highest posterior density interval (HPD) of 394–443 Ma. Selaginellaceae and Isoëtaceae separated at around 355 Ma in the Carboniferous (95% HPD: 333–382). These two families would have started their diversification at approximately 287 Ma (95% HPD: 259–317) for Selaginellaceae and, more recently, during the early Paleogene at around 25 Ma (95% HPD: 19–34) for Isoëtaceae. Lycopodiaceae started to diversify during the Cretaceous at 137 Ma (95% HPD: 105–177).

Ancestral state reconstruction showed that the three families experienced quite dissimilar evolutionary patterns in GC content and gene numbers. With the diversification of Lycopodiaceae, the GC content decreased over time, especially in *Phlegmariurus*. In contrast, the GC content of Isoëtaceae did not change greatly during diversification. Selaginellaceae showed a marked increase in GC content over time, accompanied by gene loss, whereas the number of genes in the other two families did not show a clear increasing or decreasing trend.

## 4. Discussion

### 4.1. Plastomic Diversity of Lycophytes

**Plastome Structure Divergence.** The plastomic diversity of lycophytes is reflected in two distinctive types of quadripartite structures: one is the IR structure shared by all Lycopodiaceae and Isoëtaceae species, and the other is the DR structure in most Selaginellaceae species. In-depth research on the plastome structure of Selaginellaceae shows that they include not only those with DR structures but also those in which both DR and IR structures coexist [17,21]. The DR and IR structures of the latter are reported to be dynamic, with recombination activities between repeats in plastomes giving rise to subgenomic molecules [17]. Ancestral state reconstruction analysis indicated that the DR structure was the ancestral state, whereas IR was derived in Selaginellaceae [17].

In terms of structure variation, through rearrangement analysis of each family, we found that the plastid structure of Isoëtaceae is more conservative than that of Lycopodiaceae, and much more conservative than Selaginellaceae, which is reflected in Figure 2: whole genome alignment of Isoëtaceae identified 2 locally collinear blocks (LCBs), while Lycopodiaceae identified 5, and Selaginellaceae identified 15. In previous studies, rearrangement analysis of genes also supported this conclusion [17,32]. The structural instability of Selaginellaceae plastomes is strongly associated with the DR structure and possible subgenomic molecules [17]. Most species of Selaginellaceae, or ancestor of Selaginellaceae, do not have an IR structure; however, it is generally believed that plastomes with IR structures are more stable, and plastid rearrangements are more frequent when a large inverted repeat sequence has been lost [65].

**Base Composition and Substitution****Rate Divergence.** In terms of GC content diversity, the plastids of the lycophytes almost evolved in two opposite directions. Plastid genomes of almost all Selaginellaceae possess extraordinarily high GC contents, and the GC content of their another organelle, the mitochondrion, is also GC-rich, with the highest documented mitochondrial GC content (68%) appearing in *S. moellendorffii* [66], but there is nothing unusual about the GC content of nuclear genomes in Selaginellaceae, all of which are approximately 50% and unremarkable compared with other nuclear genomes of other green plants [5,31,32]. In contrast, the GC content of Lycopodiaceae and Isoëtaceae are relatively low throughout all ferns and lycophytes [20].

The different base substitution rates of lycophytes might reflect their plastomic diversity and are related to GC content. Overall, the base substitution rates of Isoëtaceae are slower than those of Lycopodiaceae, and those of Lycopodiaceae are slower than those of Selaginellaceae. As observed in *Arabidopsis thaliana*, mutations are neither random nor evenly distributed in the genome; mutation rates are much slower in functionally constrained regions of the genome [67], which may explain the especially low base substitution rates of rRNA genes (Figure 2), as they are functionally constrained. The nuclear genome is GC-rich in rRNA loci [68], and regions near the rRNA genes also have high GC content. High GC content may reflect the nature of the rRNA gene itself, as even in prokaryotes with generally fast mutation rates, their rRNAs are extremely conserved and GC rich [69]. Therefore, the repeat region, composed of rRNA genes and their nearby areas, exhibits the highest GC content and the lowest mutation rate in the plastome because of its extremely high conservation. Since rRNA genes are generally located in the repeat regions, two repeat regions become the most GC-rich among the four sections of every quadripartite plastome, regardless of DR or IR structure. However, the GC contents of both repeat regions and single-copy regions of Selaginellaceae are almost as high as those of repeat regions, which leads to the high overall GC content of Selaginellaceae directly.

**RNA Editing Divergence.** Phylogenetic distribution of C-to-U and U-to-C RNA editing has already been found among land plants, which is related to the origins and evolutionary trajectories of two types of RNA editing [70]. In general, C-to-U editing exists in the mitochondria and plastids of all major land plant groups, while not in closely allied green algal lineages or in marchantiid liverworts, and U-to-C editing appears only in ferns, lycophytes, and hornworts [70]. Here, we found several factors influencing the RNA editing of the lycophyte. As suggested in Figure 4, different taxonomic groups have different RNA editing types and numbers. Selaginellaceae plastomes show high levels of C-to-U RNA editing, consistent with the high occurrence of C-to-U RNA editing reported in *Selaginella* plastomes [71,72]. By contrast, we found that Lycopodiaceae mainly shows U-to-C RNA editing, and two completely different clades are present in Isoëtaceae plastomes: one shows almost all C-to-U editing, and the other shows two types coexist.

We also found an effect of mutation on RNA editing. In terms of the number of RNA-editing sites, the number of sites in Lycopodiaceae was less than that in Isoëtaceae (Figure 3). This may be related to mutations at the DNA level, which are negatively correlated with the number of RNA-editing sites. The decrease in the general mutation rate may result in an increase in the number of editing sites that are retained because edited sites could have been lost due to mutations over time, as hypothesized by Tillich et al. based on evidence from seed plants [73]. Therefore, Isoëtaceae plastomes with slower base substitution rates are more likely to retain more RNA-editing sites than Lycopodiaceae plastomes (Figure 3 and Figure 4). However, given the particularity of the high occurrence of RNA editing in both *Selaginella* plastid and mitochondrial genomes [66,71,72], this model cannot account for the RNA editing of Selaginellaceae. Instead, some site recognition factors for editing events in the *Selaginella* genome may be associated with the abundance of organelle RNA editing [74]. For example, an expanded DYW-type PPR protein family (one of the key players of RNA editing [75]) is found in the *S. moellendorffii* genome [31].

### 4.2. Phylogenetic Relationships and Divergence Times

Here, we present a robust lycophyte phylogeny yielded from a phylogenomic analysis based on 93 plastomes (Figure 1). The topology of the ML tree was highly consistent with that of the BI tree, except for disagreement on the phylogenetic position of *Dendrolycopodium* and *Lycopodium* (Figure 1). Before this study, only a few plastid markers (atpA, psbA-trnH, rbcL, rps4 and rps4-trnS, trnL and trnL-F) or a few species were analyzed in phylogeny studies of Lycopodiaceae [8,18,29]. Further phylogenetic work under global and broad sampling and sequencing of Lycopodiaceae and the lycophyte is needed.

The divergence times of the major lineages of the lycophytes were near the boundaries of the geological periods (Figure 5). The first lycophytes can appear at the end of the Silurian and the beginning of the Devonian, as shown in our results (Figure 5) and supported by fossil evidence (Baragwanathia and Asteroxylon) [76,77] and many other divergence time analyses [17,36]. The divergence of the living homosporous and heterosporous lycophytes (Selaginellaceae and Isoëtaceae) was estimated to be during the early Carboniferous and late Devonian (355 Ma, 95% HPD: 333–382), similar to the results of previous molecular phylogenetic works [17,36,78]. Since the Devonian–Carboniferous boundary is associated with the second of the “big five” mass extinction events of Earth’s history, this divergence can be speculated to be infected by the joint effection of geological and climate changes. Among the three families of the lycophytes, Selaginellaceae was the first to diversify among the lycophytes during Permian and Carboniferous (287 Ma, 95% HPD: 259–317). This was followed by divergences among subfamilies of the involved Lycopodiaceae, which were dated to be during the Cretaceous and Jurasic (137 Ma, 95% HPD: 105–177). The crown age of Isoëtaceae is estimated to be 25 Ma (95% HPD: 19–34), slightly older than the age estimated by Pereira et al. [36]. The diversification of Isoëtaceae appeared recently, with a burst of species diversified in a short period of time, implying a recent evolutionary radiation, and so on in *Phlegmariurus* and *Huperzia* in Lycopodiaceae (Figure 5). This age estimate is consistent with the relatively recent radiation reported in previous phylogeny studies of Isoëtaceae [35,36] and *Phlegmariurus* [27,79].

## 5. Conclusions

Lycophytes provide a unique opportunity to study the evolution of vascular plants. However, at present, the overall genetic diversity of all lycophytes remains poorly understood. We sequenced and assembled 14 new plastomes of Lycopodiaceae species, and combined the new sequencing data with published plastomes to construct a relatively complete phylogenetic relationship of 81 lycophytes and 12 outgroups. This study explored the plastid phylogenomics, plastomic structural diversity, and compositional variation of lycophytes by assessing differences in substitution rate, GC content, genome rearrangement, RNA editing, and gene loss based on the plastomic datasets. Among the three families, Selaginellaceae diversified first and exhibited the most variable plastome structure and high GC content, together with reduced gene numbers and RNA editing sites in their plastomes; Isoëtaceae experienced recent radiation, with the lowest base substitution rate but a considerable number of RNA editing sites retained. Lycopodiaceae is also conserved in structure and exhibits the lowest GC content but the largest number of genes. Determining the plastid phylogenomics and plastomic diversity of the lycophyte will not only deepen our understanding of the evolution of these early vascular plants but will also benefit research on all terrestrial plants. However, the evolution of lycophytes cannot be resolved using plastomes alone, and more mechanisms related to the evolution of lycophytes, such as inheritance, codon usage, and adaptive evolution, remain to be studied with more approaches and samples.

## Figures and Tables

**Figure 1 genes-13-01280-f001:**
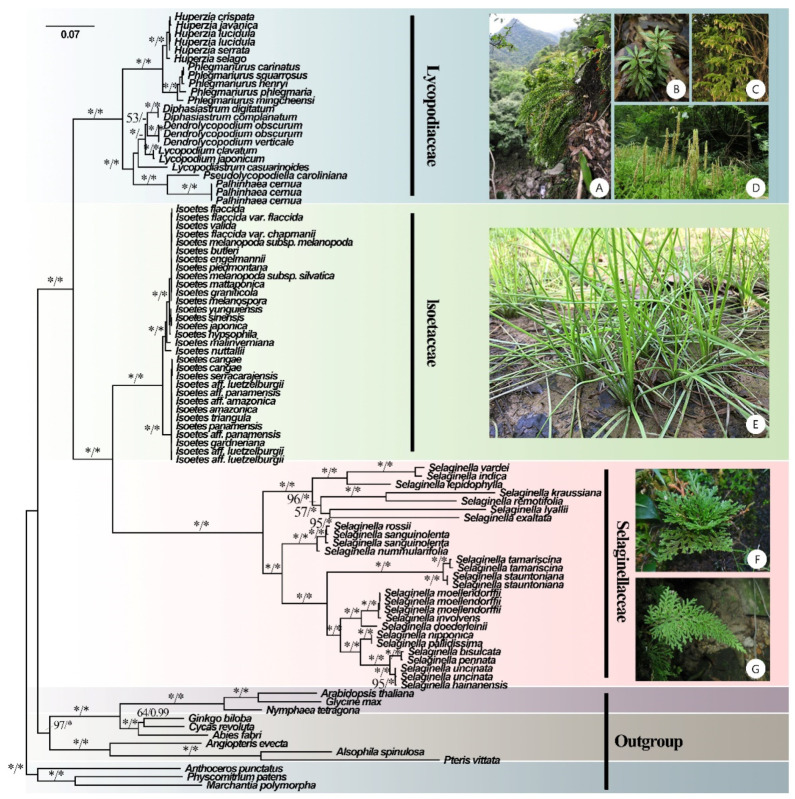
Phylogenetic tree inferred by the maximum likelihood (ML) and Bayesian methods based on 84 CDSs from 93 plastomes of 81 lycophytes and 12 outgroup species. ML bootstrap (BS) and Bayesian posterior probability (PP) support values are separated by ‘/’ and marked on each branch (* indicates BS = 100 or PP = 1). (**A**) *Phlegmariurus petiolatus* [Photos: XL Zhou], (**B**) *Huperzia javanica* [Photos: SS Chen], (**C**) Palhinhaea cernua [Photos: YH Yan], (**D**) *L. japonicum* [Photos: YH Yan], (**E**) *Isoëtes sinensis* [Photos: YF Gu], (**F**) *Selaginella tamariscina* [Photos: JP Shu], and (**G**) *Selaginella biformis* [Photos: YH Yan].

**Figure 2 genes-13-01280-f002:**
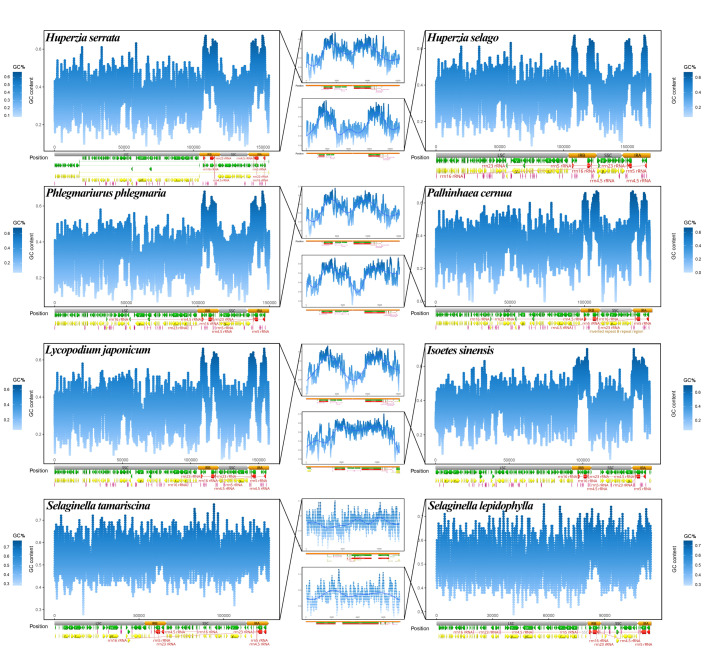
Sliding window analysis of GC content of 8 representative lycophyte plastomes. Each large figure represents the GC content distribution along a plastome. The yellow, green, red, and orange blocks at the bottom of the blue plots represent genes, CDSs, tRNA, and rRNA genes, and the two repeat regions, respectively. The middle panels represent the GC content of one of the repeat regions, connected to the corresponding species by black crossed lines. All sliding window sizes were 100 bp.

**Figure 3 genes-13-01280-f003:**
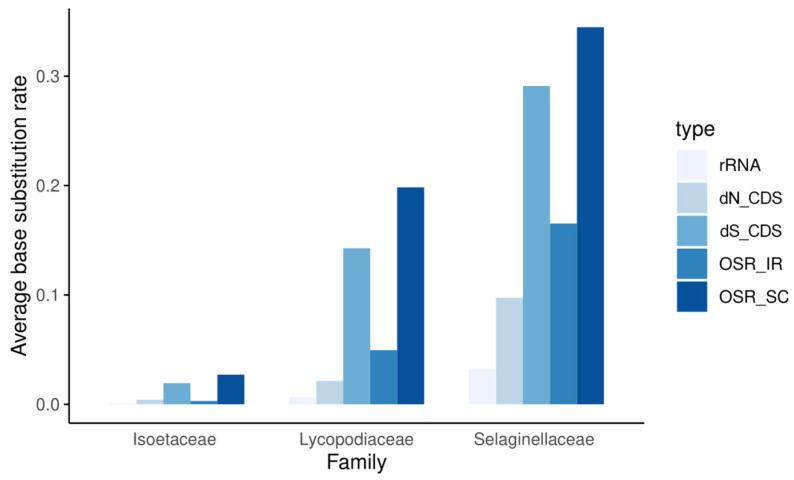
Overall substitution rates (OSRs) of noncoding sequences in repeat regions, SC regions, and rRNA regions; synonymous substitution rates (dS), nonsynonymous substitution rates (dN) of all coding sequences (CDS).

**Figure 4 genes-13-01280-f004:**
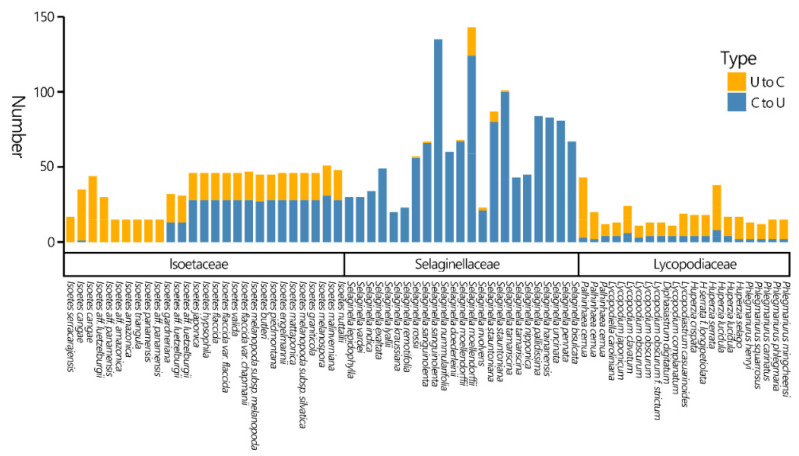
The numbers and types of predicted RNA-editing sites in the CDSs of lycophyte plastomes.

**Figure 5 genes-13-01280-f005:**
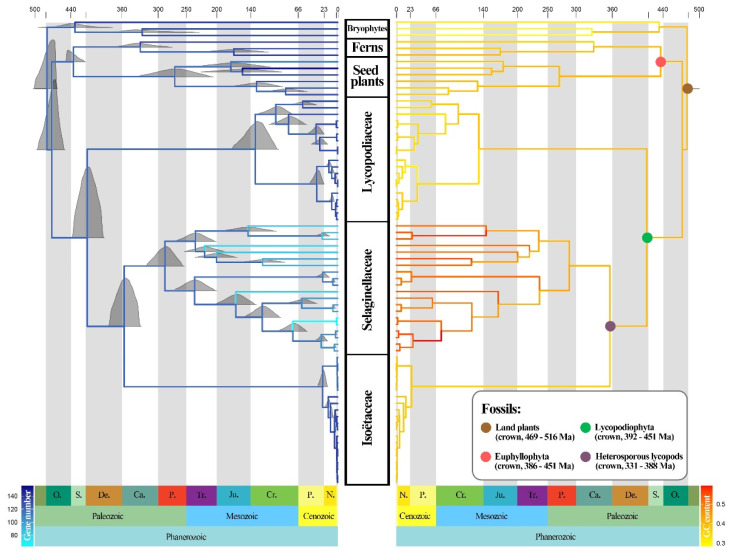
Ancestral state reconstruction of gene numbers (**left**) and GC content (**right**) along the divergence time tree of the lycophytes inferred from MCMCTree. The full posterior distributions were displayed on nodes (**left**), and the positions of fossil node calibrations were marked by colored circles (**right**).

## Data Availability

The data presented in this study are openly available in the National Center for Biotechnology Information under the accession numbers presented in Appendix A.

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
