# Peer review of "Plastid Phylogenomics and Plastomic Diversity of the Extant Lycophytes"

_genes, 2022, doi:10.3390/genes13071280_

Round 1

Reviewer 1 Report

The main aim of the paper is to analyse the plastid phylogenomics and plastomic diversity of three lycophyte families

The paper shows interesting information and well explained except some aspects in material and methods and the results

Specific comments

Line 48. More explanation about their function is needed.

Line 114. This information should be given in a graph or a table to have all the information available and easy to see.

Line 178. The yellow, green, red and orange bottom is unreadable.

Line 342. Where those individuals come from? Just one individual per species? No variation within species expected? A list of the samples and their information is needed. Table S1 shows new sequenced samples in bold, but they cannot be distinguished clearly.

Author Response

Thanks for your constructive comments. We have carefully revised the manuscript according to your suggestion.

Comment 1: Line 48. More explanation about their function is needed.

Response: Thanks for your advice! We have added more description of their function already.

Comment 2: Line 114. This information should be given in a graph or a table to have all the information available and easy to see.

Response: Thanks for your suggestion. We have already listed all the information in Table S1, but taken your advice into consideration, we added a graph (Figure S1) to make all the information easy to see.

Comment 3: Line 178. The yellow, green, red and orange bottom is unreadable.

Response: Thank you for your suggestion. Since there are too many items at the bottom, it is a little bit difficult to display them clearly. But we have adjusted the size and resolution of the image, and enlarged the yellow, green, red and orange bottom as much as possible. If you open the Figure 1 file and enlarge it, they can be seen clearly.

Comment 4: Line 342. Where those individuals come from? Just one individual per species? No variation within species expected? A list of the samples and their information is needed. Table S1 shows new sequenced samples in bold, but they cannot be distinguished clearly.

Response: Thank you for your comments. We add more information about these individuals in the manuscript. They are one individual per species and no variation within species, because we are more interested in changes at the family and genus levels than at the species level. We are sorry that Table S1 can not be distinguished clearly, so we marked and annotated them in Table S1. Hope this clears up your confusion. Thanks again. 

Reviewer 2 Report

The manuscript reads very well and the text is very easy to follow. The authors have cited most references considering the scope of this study, and have clearly defined its aims. I have only the following minor comments, which I think might be useful to this article:

- The text in the figures is very hard to see. Except for Figure 3, the size of the fonts in the remaining ones is very hard to follow. For instance, the authors refer to "highly supported tree topologies" in Figure 1 but I cannot see this. Perhaps enhance the font in this Figure or use different lines to indicate different BS values? Similar issues occur in the remaining figures.

- In relation to the species indicated in taxon sampling, abbreviations concerning the genus should only have one letter, e.g., D instead of De (and so on).

- Results are sometimes mixed with discussion, and even citations are included in the results. I would avoid this but I leave the option to the authors.

- About Data Availability: thank you for submitting it to NCBI. I have seen the new data in Table S1 but it is not (yet) available in NCBI. Please remember to do so, when the article is accepted. 

Author Response

Thanks for your constructive comments. We have carefully revised the manuscript according to your suggestion.

The manuscript reads very well and the text is very easy to follow. The authors have cited most references considering the scope of this study, and have clearly defined its aims. I have only the following minor comments, which I think might be useful to this article:

Comment 1: The text in the figures is very hard to see. Except for Figure 3, the size of the fonts in the remaining ones is very hard to follow. For instance, the authors refer to "highly supported tree topologies" in Figure 1 but I cannot see this. Perhaps enhance the font in this Figure or use different lines to indicate different BS values? Similar issues occur in the remaining figures.

Response: Thank you for underlining this deficiency. We have enhanced the size of the fonts in Figure 1, 2, and 5 in the revised manuscript. Hope this clears up your confusion.

Comment 2:  In relation to the species indicated in taxon sampling, abbreviations concerning the genus should only have one letter, e.g., D instead of De (and so on).

Response: Thanks for your suggestion, we used abbreviations to distinguish these genera with the same first letter, but taken your advice into consideration, we use the whole genus names to avoid confusion.

Comment 3:  Results are sometimes mixed with discussion, and even citations are included in the results. I would avoid this but I leave the option to the authors.

Response: Thank you for underlining this deficiency. We have revised the manuscript to avoid these discussions and citations in the results, especially in section 3.2.

Comment 4:  About Data Availability: thank you for submitting it to NCBI. I have seen the new data in Table S1 but it is not (yet) available in NCBI. Please remember to do so, when the article is accepted. 

Response: Thanks for your reminding. We have already uploaded them to NCBI, but they are set to be available until next year. We will try our best to make it available in NCBI when the article is accepted. Thanks again.

This manuscript is a resubmission of an earlier submission. The following is a list of the peer review reports and author responses from that submission.